# Landscape Pattern and Ecological Risk Assessment in Guilin Based on Land Use Change

**DOI:** 10.3390/ijerph20032045

**Published:** 2023-01-22

**Authors:** Yanping Lan, Jianjun Chen, Yanping Yang, Ming Ling, Haotian You, Xiaowen Han

**Affiliations:** 1College of Geomatics and Geoinformation, Guilin University of Technology, Guilin 541004, China; 2Guangxi Key Laboratory of Spatial Information and Geomatics, Guilin University of Technology, Guilin 541004, China

**Keywords:** landscape vulnerability index, landscape disturbance index, ecological risk assessment, spatial correlation, Guilin city

## Abstract

The land use and ecological risk patterns in Guilin, which is the only innovation demonstration zone under the National Sustainable Development Agenda in China with a focus on the sustainable use of natural resources, have changed significantly as a result of the combined impact of climate change and human activities, thus presenting challenges to the sustainable development of the local area. This research employs an ecological risk assessment model and spatial analysis techniques in order to analyze the spatial correlation between land use and ecological risk, and to evaluate the spatial and temporal evolution characteristics of ecological risk at the overall and county scales in Guilin. The results reveal the following: (1) A total of 1848.6 km^2^ land types in Guilin have changed from 2000 to 2020, and construction land has gradually expanded from the central urban area to the suburbs with increasing internal stability each year. (2) The ecological risk level in Guilin showed a decreasing trend at the city scale, but some regions still showed an increasing trend at the county distribution scale. (3) The ecological risk value in Guilin has significant spatial correlation, and the spatial distribution showed a clustering effect, which was consistent with the spatial distribution of ecological risk class areas. The research results can provide a reference for ecological risk control and sustainable development of landscape resource cities.

## 1. Introduction

In recent years, the rapid growth of urbanization and industrialization has led to irrational land use and development activities that pose significant risks to the ecosystem, including accelerated soil erosion [1], the loss of biodiversity [2], and a decline in the ecological land area [3]. Consequently, in order to prevent the impact of natural and human factors on ecosystems, it is necessary to develop an ecosystem risk assessment system through the investigation of the spatio-temporal evolution of the risk characteristics of regional ecosystems [4,5]. Ecological risk assessment involves the evaluation of the likelihood and severity of adverse effects of external factors on the functions and structure of an ecosystem [6]. In addition, it can also serve as a crucial technical means for exploring regional ecological sustainable development. In traditional ecological risk assessment techniques, only a single risk source and risk receptor are taken into consideration [7], both of which are susceptible to influences from external factors. Accordingly, scholars have expanded the scope of assessment to also encompass the landscape level in order to reduce the effects of the uncertainty and complexity of changes within an ecosystem on the risk assessment process.

As a basic unit of human activities, the landscape is characterized by spatial heterogeneity [8], which is closely related to the resilience [9], stability, and diversity of the ecosystem [10], as well as a number of other indicators. Landscape ecological risk refers to the likelihood that changes in the landscape and environment may adversely affect ecosystems [11,12]. It is both a reflection of landscape heterogeneity, and the result of different scales of disturbance and ecological processes [13,14]. The landscape ecological risk assessment model differs from the traditional ecological risk assessment models; it emphasizes the spatio-temporal heterogeneity, and the evolution of risk characteristics. It also takes the spatio-temporal evolution of ecological risk into account, and is more suitable for the assessment of ecological risks over a longer period of time [15]. The risk source-sink model and the landscape pattern model are two mature and commonly used methods for the assessment of landscape ecological risks. The research methods based on the source-sink model are more constrained, focusing primarily on potential risk source materials as well as the analysis of receptors [16], while the landscape pattern model can assess the ecological change of the entire region, and establish a link between landscape structure and phenomena based on the landscape index [17]. Currently, the ecological risk assessment model based on the landscape patterns is focused on the quantitative evaluation of ecological risks from the perspective of the spatial landscape patterns caused by changes in land use intensity and land use type [18,19].

Land use is directly related to natural elements and human actions. Land use change may directly impact the risk to the regional landscape ecosystem, in addition to affecting and changing the structure and use of land resources [20]. Landscape ecological risk assessment of land use change is a regional ecological risk assessment method based on the perspective of spatial pattern. The method is based on the theory related to pattern process, which can directly reflect the ecological risk in the structure and composition of landscape pattern [3]. In recent years, the ecological risk assessment method based on the changes in land use has become the focus of research throughout the world, and ecological risk research plays an important role in urban planning [21], resource allocation [22], and ecosystem protection [23]. For example, Potter et al. [24], based on the landscape pattern model, investigated the ecological risk associated with non-point source pollution in North Carolina, USA. Liu et al. [25] used landscape indices and pattern analysis methods, examined the influence of the roads on the landscape in the Lancang River Basin, thereby assessing the severity of the impact of different types of roads on regional ecological risks. However, in terms of evaluation areas, the current ecological risk assessment areas are mainly focused areas such as coastal watersheds [26], urban administrative regions [27], and ecologically fragile source areas [28], and the research on ecological risk assessment of landscape resource cities is not deep and complete. In terms of evaluation scales, the existing ecological risk assessment research scales mainly focus on the provincial and municipal scales [29], and it is easy to ignore the spatial and temporal evolution of local ecological risks in the evaluation process, which leads to the lack of sufficient environmental protection awareness in the planning process of counties and districts, and reduces the practicability of ecological risk assessment results [30]. Therefore, it is practical and realistic to conduct ecological risk assessment research on landscape resource cities, to comprehensively analyze the spatial evolution characteristics of local ecological risks at the county scale, and to provide ecological protection suggestions.

The city of Guilin in China is a typical characteristic of a karst area with undulating terrain, frequent natural disasters, and a fragile ecological setting. As a demonstration region for China’s Sustainable Development Agenda, Guilin actively carries out the work of returning farmland to forest, and controlling the rocky desertification of the land to maintain the stability of the ecosystem. However, the type and intensity of land use in Guilin has undergone drastic changes due to the rapid advancement of urbanization and industrialization, which has led to the loss of soil and water, and a significant increase in the regional ecological risks [31]. The majority of research on land use change in Guilin is currently primarily conducted from the perspective of carbon storage, land change mechanisms, ecological security patterns, etc., while research on ecological risks resulting from changes in landscape patterns due to land use change is comparatively rare [20]. In light of this, this study takes Guilin as its research area (Figure 1), then based on the land use data collected in 2000, 2010, and 2020, we analyze the spatial and temporal distribution characteristics of ecological risks in the process of urbanization, and then we discuss the change principles of ecological risks at the county scale in order to provide reasonable reference suggestions for Guilin land use planning and management.

## 2. Materials and Methods

### 2.1. Study Area

The city of Guilin is situated in the northeast of the Guangxi Zhuang Autonomous Region (109°36’50”E–111°29’30”E, 24°15’23”N–26°23’30”N). This city is located at the intersection of the provinces of Guangxi, Hunan, and Guizhou. The entire administrative territory of Guilin measures 27,800 square kilometers, divided into six municipal districts and eleven counties. It makes up 11.74% of the total land area of the Guangxi region (Figure 1). Guilin is located at a low latitude and is characterized by a mild subtropical monsoon climate, with an annual average temperature of over 19 °C, and abundant rainfall with an annual average rainfall of 1887.6 mm [32]. The research area consists of a typical karst landform region with high topography to the west, north, and southeast, as well as flat ground in the center [33]. The city of Guilin is well-known internationally for its ecotourism attractions, and is home to four national nature reserves and seven nature reserves at the autonomous region level. However, due to the particularity of geographical location and landform, Guilin is in short supply of available land resources, and the city is confronted with a variety of ecological and environmental problems, such as soil erosion, karst collapse, and the rocky desertification of arable land [34]. In addition, due to the unreasonable planning of new development areas and the massive construction of transportation corridors, the connectivity of the urban landscape ecosystem has received a great impact, and the sustainable development of Guilin is facing a serious challenge.

### 2.2. Data Source

The land use data for this study were obtained from the Data Center for Resources and Environmental Sciences and the Chinese Academy of Sciences (RESDC) (https://www.resdc.cn/ (accessed on 10 November 2022)). These data were obtained by visual interpretation of Landsat TM/ETM and Landsat OLI remote sensing image data, with a resolution of 30 m [35]. In order to better explore the temporal and spatial change characteristics of ecological risks, according to the actual development of Guilin, land use data in 2000, 2010, and 2020 were selected. The land-use types included six first-level land classes and twenty-five second-level land classes. The overall classification accuracy of the Kappa coefficient is more than 85%, making it the most accurate land-use remote sensing product in China. The results of relevant research have shown that the land use data are time-sensitive and easily available, and can be used for studies related to land use change [5]. These data also play an important role in the investigation of land resources, as well as ecological protection research [26]. In this study, the dataset was reclassified into six land-use types: arable land, woodland, grassland, waters, construction land, and unused land (Figure 2). 

### 2.3. Research Methods

#### 2.3.1. Land-Use Change

The reclassified land-use data collected in the years 2000, 2010, and 2020 were spatially calculated using GIS software in order to derive the area, and the dynamic degree of changes of each land’s use. In addition, in the case of the land use data collected from different periods, the land use transfer matrix of Guilin was obtained through the method of superposition calculation. The specific experimental process is shown in Figure 3.

#### 2.3.2. Landscape Ecological Risk Model

Landscape ecological risk assessment is measured by the external disturbance intensity and internal vulnerability. Moreover, it mainly includes the three primary aspects: the choice of the landscape pattern index, the division of the ecological risk plot, and the creation of the landscape ecological risk model [36,37]. In order to better synthesize the land use situation of karst landform in Guilin, and quantitatively represent the degree of ecological risk in risk units, five landscape pattern indices, including landscape fragmentation index (*C_i_*), landscape separateness index (*B_i_*), landscape dominance (*D_i_*) index, landscape disturbance index (*N_i_*), and landscape vulnerability index (*O_i_*) were selected for this research and analysis [38]. The magnitude of each landscape pattern index was calculated by Fragstats 4.2 software on land use data. The specific index formula and its significance are shown in Table 1.

It has been demonstrated that in cases where the sample square is approximately 2 to 5 times greater than the average area of the landscape patches in the study area, the sample is capable of accurately reflecting all of the information about the landscape pattern of the surrounding sampling site [39,40]. In this study, the landscape heterogeneity and patch size of Guilin were comprehensively considered. We divided Guilin into 5 km × 5 km grid units by the method of equal spacing, which resulted in a total of 1229 ecological risk units. In each evaluation unit, the ecological risk index (ERI) is calculated, and the results are assigned to the central pixel of the evaluation area [15]. Landscape pattern index can not only reflect the extent of regional land use, but also has a certain correlation and connectivity with the spatial evolution of ecological risk [41,42]. Therefore, in order to better reflect the relationship between landscape index and ecological risk, this study used landscape disturbance index and landscape vulnerability index to establish the comprehensive ecological risk index of Guilin (ERI) (Formula 1). Finally, Kriging interpolation was carried out using the ArcGIS software module to carry out ball fitting for ecological risk map values. The natural breakpoint method in ArcGIS was used to divide the ecological risks in Guilin area from 2000 to 2020 into five levels: lowest (ERI ≤ 0.120), lower (0.120 < ERI ≤ 0.125), medium (0.125 < ERI ≤ 0.130), higher (0.130 < ERI ≤ 0.137), and highest (ERI > 0.137).
(1)$$ERI_{k}=\sum_{i\;=\;1}^{n}\frac{A_{k_{i}}}{A_{k}}\sqrt{N_{i}\times O_{{}_{i}}}$$
where *ERI_k_* is the ecological risk index of the *k*-th risk plot; *k* is the sampling unit; *A_ki_* is the area of the landscape *i* in the *k* risk zone; *A_k_* is the total area of all the landscape risk of the sampling unit; *n* is the number of land use types; *N_i_* is the landscape disturbance index of *i*; *O_i_* is the landscape vulnerability index of *i*.

#### 2.3.3. Spatial Correlation Analysis Method

Spatial statistical analysis can quantitatively express the spatial correlation of variables. It can be used to explore the spatial pattern and distribution characteristics of regional ecological risks, which can be divided into global spatial autocorrelation analysis and local spatial autocorrelation analysis [48]. In addition, it helps to understand the interdependence between a variable and data from the same distribution area, or the extent to which its distribution characteristics affect its neighbors [49]. GeoDa software is commonly used for correlation analysis.

Global spatial autocorrelation is primarily used to determine whether a phenomenon contains aggregation properties in space [50], and Moran’s *I* is commonly used to represent its correlation. The Moran’s I index (*I^i^*) has a value that falls between [−1, 1]. In general, the strength of a phenomenon’s spatial autocorrelation increases as the value approaches −1 or 1. When *I* < 0, it indicates that the spatial correlation is negative; when *I* > 0, it means that the spatial correlation is positive; when *I* = 0, it indicates that the phenomenon is not spatially correlated and is randomly distributed. The global autocorrelation is calculated as in the following formula:(2)$$I^{i}=\frac{k\sum_{a\;=\;1}^{k}\sum_{b\;=\;1}^{k}w_{ab}(y_{a}-\bar{y})(y_{b}-\bar{y})}{\sum_{a\;=\;1}^{k}\sum_{b\;=\;1}^{k}w_{ab}\sum_{a\;=\;1}^{k}{\left(y_{a}-\bar{y}\right)}^{2}}$$
where *k* indicates the total number of sampling areas; *y_a_* and *y_b_* are the attribute values of *a* and *b*, respectively; *y* is the sample mean value; and *w_ab_* is the spatial weight.

In cases where a phenomenon is anomalously distributed and is not correlated, it is possible to better capture the spatial aggregation distribution pattern and the significance of aggregation between a region and its neighboring regions by employing local autocorrelation analysis [51]. Visual representation of the spatial distribution of local differences using LISA aggregation maps, with five main representations: Not-significant aggregation (NS), High-High aggregation (HH), Low-Low aggregation (LL), Low-High aggregation (LH), and High-Low aggregation (HL) [39]. The calculation formula is as follows:(3)$$I^{j}=x^{\prime }{}_{a}\sum_{a\;=\;1}^{k}w_{ab}x^{\prime }{}_{b}$$
where *I^j^* is the spatial unit value of the LISA index; *x_a_* indicates the sample unit standards of *a*; *x_b_* indicates the sample unit standards of *b*; and *w_ab_* is the spatial weight.

## 3. Results

### 3.1. Spatio-Temporal Characteristics of Land Use Change

The trends of each land use type in Guilin from 2000 to 2020 are shown in Figure 2 and Table 2. On the whole, woodland, grassland, and arable land were the main land types in Guilin, which together account for over 90% of the total area of the research area. Most of the woodland areas were located in the mountainous regions of the northwest and east, while construction land and arable land were primarily located in the central, northeastern, and southern regions. In the past 20 years, there was a significant expansion in the total land mass of construction land, and the expansion occurred in areas including Yongfu Country, Lingui District, Quanzhou Country, Yangshuo Country, Lipu Country, Pingle Country, Yanshan District, and Diecai District. During the study period, a number of changes in land use occurred to different degrees. The total mass of arable land decreased by 94.50 km^2^, the total land mass of woodland decreased by 54.23 km^2^, and the land mass of grassland decreased by 50.04 km^2^. The total land mass of waters and construction land increased by 27.76 km^2^ and 167.69 km^2^, respectively, while there was little change in the land mass of unused land.

According to the land use type transfer matrix in Guilin between 2000 and 2010 (Table 3), during the study period, the area of land use change in the study area was approximately 222.62 km^2^, and the main transfer type was arable land. Additionally, the transfer area of arable land was 74.85 km^2^, accounting for 33.62% of the total change area, and the area transferred from woodland and grassland was 66.35 km^2^ (29.80%) and 63.98 km^2^ (28.74%), respectively. The three primary categories of the land shift were arable land to woodland, woodland to grassland, and grassland to woodland, which together accounted for 13.85%, 15.93%, and 20.41% of the total area of land change. The increase in unused land and construction land was primarily caused by a decrease in arable land.

According to the land use transfer matrix for the years 2010 to 2020 (Table 4), land use areas and types have undergone significant changes during this decade. The changing area from 2010 to 2020 was 1625.98 km^2^, about 7.3 times that of 2000 to 2010, and the fluctuation of land use intensity was more noticeable. Among these change areas, arable land and woodland comprised the largest area of change, 538.97 km^2^ and 621.14 km^2^, respectively, accounting for approximately 33.15% and 38.20% of the overall land use change area. Arable land was primarily transformed into woodland and construction land, and woodland was primarily transformed into arable land and grassland. There was a significant expansion in construction land during this time period. Except for the two types of land use of waters and unused land, which were not transformed into construction land, their remaining land use types underwent different areas of conversion. Construction land grew by 116.69 km^2^, which accounts for approximately 12.84% of the total land area that was altered.

### 3.2. Spatio-Temporal Characteristics of Land Use Change

The intensity of inter-conversion between land use categories had a significant increase between 2000 and 2020, which resulted in a substantial change in the landscape pattern index in Guilin (Table 5). The number of patches in waters, construction land, and arable land indicates a decreasing trend; however, there was an increase in the number of patches across other land-use categories. Construction land was highly fragmented, exhibiting a clear declining trend. The separateness index of woodland was the smallest, indicating a prominent feature, whereas the separation index of unused land was the largest, followed by construction land. The dominance index of arable land and woodland was high, the two most common forms of landscape within the research area; however, they exhibited a declining trend. Arable land, grassland, and waters all experienced a rise in the disturbance index over time, while woodland, construction land, and unused land exhibited the reverse trend. The vulnerability of the six land use types can be ranked as follows: unused land > waters > arable land > grassland > woodland > construction land.

### 3.3. Correlation Analysis of Landscape Ecological Risk

The highest ecological risk index of Guilin increased from 0.169 in 2000 to 0.170 in 2020, and the risk degree increased by 0.59%. The average ecological risk indexes in 2000, 2010, and 2020 were 0.123, 0.121, and 0.121, respectively, demonstrating only a slight overall difference. Between 2000 and 2020, the ecological risk areas mainly included lowest-risk, lower-risk, and medium-risk (Figure 4), accounting for 48.68%, 18.73%, and 12.88% of the total land mass of the study area on average, whereas the higher-risk and highest-risk areas accounted for proportionally smaller areas of 17.48% and 2.22% of the total area, respectively (Figure 5). The lowest ecological risk areas within the research area were primarily distributed in the northwest and eastern regions, while the remaining risky areas were distributed in the northeast, central, and southern regions. During the 2000 to 2020 period, the proportion of lowest-risk and highest-risk areas showed a decreasing trend, specifically by 0.69% and 1.58%, respectively. Over time, more areas were classified as lower-risk or medium-risk, with the medium-risk areas rising by 1.58%, while the proportion of higher-risk areas decreased first and then increased. The overall ecological threat to Guilin has decreased during the study period, and the ecological situation has steadily improved.

According to Figure 6, it was indicated that the lowest-risk areas between the 2000 and 2020 time period were distributed primarily in woodland and grassland areas, which accounted for a higher proportion of the area of these two land use types, and the area change trend was small during that period. The proportion of woodland and grassland area in the highest-risk areas was lower, at 1% and 2%, respectively. Arable land and waters were distributed primarily in the higher-risk areas, and the proportion of the land mass of these two features showed a decreasing trend during the 2000 to 2020 time period, decreasing by 4% and 8%, respectively. However, other ecological risk areas exhibited an even distribution phenomenon. Construction land was distributed primarily in the highest-risk and higher-risk areas, which represented 31% and 41% of the total construction land area in 2000, and 29% and 43% of all construction land area in 2010. In contrast, by 2020, the proportion of the highest-risk areas had declined significantly, representing only 4% of the total area of construction land, and higher-risk areas had a small increase, accounting for 61% of the total construction land area. In 2000, unused land was distributed primarily in the highest-risk areas, lower-risk areas, and lowest-risk areas. However, the unused land was distributed in the higher-risk areas in 2010 and 2020. During the 20-year period, the ecological risk level of unused land decreased in 2020, and the highest-risk area represented 38% of the total area of unused land, and the ecological internal structure was stable.

Figure 7 showed the distribution area of ecological risk plots in each district and county of Guilin from 2000 to 2020. According to the figure, it was shown that there was a slight variation between the lowest-risk and lower-risk zones in each district and county, except for Deicai District, Xiangshan District, Xiufeng District, Yanshan District, and Qixing District, which had no distribution; all other districts and counties had distribution, and the areas had become smaller in size. Quanzhou County had the highest area of medium-risk zones between the years 2000 and 2020, accounting for 27.68%, 27.61%, and 28.95% of the area of medium-risk areas, respectively, which indicates a slight upward trend. The higher-risk area was not distributed in Longsheng County, but distributed in other districts and counties. In 2000, the highest-risk areas were distributed in Xiufeng District, Diecai District, Xiangshan District, Qixing District, Yanshan District, Yangshuo County, Lingui County, Quanzhou County, Xing’an County, Yongfu County, Guuangyang County, Pingle County, and Lipu County, of which Lingui District seemed to have the highest degree of distribution, accounting for 19.45% of the total distribution area. In 2010, the distribution of the highest-risk areas decreased from that of 2000 in Lipu County, while the areas increased in Lingchuan County. As compared with 2000 and 2010, the highest-risk area appeared substantially less in 2020, among which there was no distribution of this graded area in Xiufeng District, Gongcheng County, Longsheng County, Lipu County, and Resource County. Compared with 2010, Quanzhou County and Xing’an County were the most widely distributed areas of highest-risk, with an increase of 10.92% and 8.39% in 2020, while the areas of other counties exhibited a decreasing trend.

### 3.4. Spatial Autocorrelation Analysis

According to the global spatial autocorrelation map (Figure 8), Moran’s *I* index in the years 2000, 2010, and 2020 exhibited positive results, which were greater than 0.05; they were 0.6124, 0.6035, and 0.5856, respectively. This suggested that the ecological risk index in the study area exhibited a strong autocorrelation and that adjacent plots affected each other. In the study area, there was a weakening aggregation trend in ERI from 2000 to 2020, and the aggregation of ERI decreased in general. It was mainly related to the change in land use intensity, indicating that the spatial aggregation and spatial differentiation of ecological risks decreased gradually with the increase in time.

During the study period, the spatial morphological aggregation in Guilin was mainly characterized by not-significant aggregation, Low-Low aggregation, and High-High aggregation (Figure 9), which was characterized by “homogenous aggregation and heterogeneous separation”. The Low-High aggregation and High-Low aggregation were more scattered in terms of distribution, and were mainly distributed around High-High aggregation. Lingui District, Yanshan District, Xiangshan District, Xiufeng District, Diicai District, Lingchuan County, Yongfu County, Yangshuo County, Pingle County, Lipu County, Gongcheng Yao Autonomous County, and the central urban areas of Quanzhou County were the main distribution areas for the High-High aggregation. These locations were mostly composed of arable land and construction land, with a large proportion of construction land. The distribution of Low-Low aggregation was primarily in the northwestern, southern, and eastern regions, and the landscape type was dominated by woodland and grassland. The number of Low-Low aggregation types exhibited a declining trend; not-significant aggregation, High-High aggregation, and High-Low aggregation types showed a rising trend; and Low-High aggregation types showed a rising and then a declining trend, as shown in Table 6.

## 4. Discussion

### 4.1. Landscape Characteristics and Land Use Change Analysis

In the early 21st century, in the absence of rational urban planning, industrial land and construction land in cities were scattered, leading to inefficient utilization of urban land and fragmentation of spatial distribution in China [52]. As a result of the rapid development of the economy, the development mode of Guilin has changed dramatically, which has significantly impacted the regional land use structure [50,53]. Guilin is a typical karst city with many mountains and steep slopes. It is distinguished by a low utilization rate of arable land, low proportion of transportation land, and small scale of industrial land, as well as township land [54]. In addition, as a result of natural disasters, socioeconomic developments, and a number of other factors, arable land in Guilin has been gradually replaced by construction land, leading to an increase in fragmentation and the occurrence of stone desertification and soil erosion in arable land. Consistent with other studies of the same period [55,56], different land use types have different forms of change due to their different distribution characteristics and functions. In the Guilin area, vast agricultural land and locations along transportation corridors adjacent to urban centers and strategic areas are used for real estate development, commercial development zones, and industrial parks [54]. A few examples of this are Lingui District, Qixing District, and Quanzhou County. The majority of construction land is located in the central, northwestern, and southern regions. These areas have gradually transitioned from having a dispersed distribution to having a concentrated and continuous distribution, forming a strong agglomeration and stability. In part, this suggests that the integrated use of land will increase as the city grows. Among these areas, the eastern part of Yongfu County, Lingui District, Quanzhou County, and the southern part of Pingle County were primarily agricultural and industrial areas where arable land and construction land were intertwined with development land, and they also exhibit a high degree of land use intensity. The woodland was primarily distributed in Longsheng County, northern Lingui District, northern Lingchuan County, Yongfu County, Ziyuan County, and western Guanyang County, among other places. With a huge distribution area and numerous patches, the dominance of the landscape is greater and less influenced by human activity. In addition, industrialization has been slow in these areas, which are dominated by agriculture and animal husbandry.

The development trend of the whole land use in Guilin is closely related to the policies that have been implemented by the government [57]. In 2008, the Guilin Municipal Government published the “Outline of Guilin City Land Use Master Plan”, which had a significant influence on the land use reform. During the first phase of the reform (2006–2010), the arable land area of Guilin declined sharply, while there was no discernible trend toward urban expansion. This is primarily due to the development of agricultural land adjacent to the original urban centers, and major traffic routes into industrial parks and real estate during this period. As a result, the ecological makeup of the land was destroyed, land use efficiency was decreased, and the pattern of the urban landscape became fragmented. Additionally, since 2010, the government has placed an emphasis on land zoning development, changing dramatically the land structure of Guilin. The change in land supply system not only promoted the rational expansion of urban space, but also made the agglomeration pattern of urban patches more concentrated [58]. In addition, Guilin has also carried out environmental restoration and protection of major tourist attractions, and has focused on the development of industrial land. With the improvements in the land supply patterns and the farmland protection policies, land usage in Guilin has become more standardized and effective.

### 4.2. Analyses of Spatio-Temporal Ecological Risk Evolution

In areas where human activity is frequent, the spatiotemporal changes in landscape patterns and ecosystem risks provide an accurate reflection of the effects of human activity on natural ecosystems [21,59]. The ecological risks in Guilin were mainly lowest, lower, and medium risk, and the level of ecological risks showed a decreasing trend. There was a steady decline in the area with the highest-risk grade, as well as an improvement in the ecological environment. Due to the influence of karst topography, the land use pattern in Guilin had obvious regional differences [7]. Arable land and construction land were mainly distributed in the central, southern, and northeastern parts of the study area, while wood land was mainly concentrated in the western part. Due to the rapid urbanization, farmland and grassland were continuously transformed into urban land, and the spatial distribution of ecological risk grade areas in Guilin showed obvious territoriality. The majority of the highest-risk areas were concentrated in areas with flat terrain and high human activity, particularly in some urban construction areas such as the southern portion of Lingchuan County, Diecai District, the eastern half of Lingui District, and the eastern part of Xing’an County. In the hilly regions with dense vegetation cover, the risk zones were fewer, the internal structure of the landscape type was more stable, and there was less anthropogenic disturbance, such as in Longsheng County, northern Xing’an County, northern Lingui District, and a number of other areas.

In the year 2000, Guilin was in a state of backward development, with a slow rate of urbanization and a predominantly conventional tourism economy [60]. In this period, the urban construction land area was small, and the plot ratio was low, and urban expansion mainly sacrificed farmland and grassland. In 2000, the ecological risk of the landscape was high due to the reckless encroachment and destruction of the semi-natural landscape by human activities, and the fragmentation of the urban landscape was severe. Combined with the information of Guilin Statistical Yearbook from 2000 to 2020 [48], it can be summarized that the central and northeast regions of Guilin had higher GDP and population density, rapid development of urbanization level, and the development mode was transitioning from the primary industry in the central city to the local industry. From 2000 to 2010, the area with the highest ecological risks decreased significantly, indicating a general improvement in the ecological environment during this time period. There has been a decrease in the highest-risk area in the downtown and the surrounding areas between 2010 and 2020, but there has been an increase in the northeast. Moreover, the area of medium and higher ecological risk zones displayed an upward trend, indicating that even though the ecological surroundings have improved over time, there were still potential ecological problems in some areas due to the rapid development of local industries. As compared with 2000, in 2010 and 2020, the areas of highest ecological risk in the six districts (Lingui District, Xiangshan District, Xiufeng District, Qixing District, Deicai District, Yanshan District) changed significantly, and exhibited a trend of decline. The land use activities in these places were possibly focused in older urban areas, the land use types remained constant, while the quantity of landscape patches increased, and the landscape as a whole became more stable.

Different from previous studies [61,62], the research period selected in this study was mainly related to the land-use planning policies of Guilin. The key time nodes were selected to study the evolution of ecological risks, and the results obtained were consistent with the actual development of Guilin, which can more profoundly reveal the ecological risk characteristics of Guilin. However, this research only evaluated the spatial and temporal evolution of ecological risks in Guilin from the perspective of land use data and landscape pattern index, and did not explore the correlation between the spatial distribution of ecological risks and landform in karst areas in combination with other ecological factors, and the evaluation indicators and criteria were relatively simple. Ecological risk assessment is a complex project with a wide scope, and future research needs not only to quantitatively assess the spatial and temporal evolution of ecological risks, but also to conduct an in-depth investigation of the intrinsic evolution mechanism of ecological risks by special topography and landscape, so as to provide reasonable suggestions for the sustainable use of landscape resources.

### 4.3. Suggestions on Ecological Risk Prevention and Control

Guilin City is a new industry based international tourist city situated at the southern end of the Xiang–Guangxi corridor. Due to the rapid expansion of urbanization and irrational land use since the early 21st century, the natural environment of Guilin has become more fragmented [63]. In light of the findings of this study and the actual needs of Guilin, the following recommendations are made:

(1) Land use supervision and ecological restoration should be reinforced for medium and highest-risk areas, such as Quanzhou County, Lingui District, and Xing’an County. It is imperative to implement space control in the development of urban construction land, and land use mechanisms should be strengthened. Additionally, it is crucial to strengthen ecological security monitoring in these areas, and give priority to protection, management, and ecological projects in key areas. 

(2) In the lowest and lower risk areas, conservation of existing landscape types should be enhanced, and reasonable land use changes should be encouraged. In accordance with the distribution area of each risk level of a district, the county level, district level, and township level should be divided into general control areas, strict control areas, and important control areas, and districts should be managed according to local conditions. Regulation and prevention policies must be implemented by the general control area in order to maintain the diversity of landscape types; strict control areas must implement the land planning systems and carry out reasonable urbanization construction; most of the important control areas should be concentrated in urban construction areas, and therefore environmental restoration should be improved, and more investments should be made in urban greening construction to maintain its ecological benefits.

(3) For industrial regions, particularly township businesses, it is essential to implement centralized planning and development, create high-tech industrial parks and industrial concentration areas, and encourage the intensive use of the industrial property. The environmental protection and land reform guidelines, established by the state and the federal government, must be strictly followed, and attention must be paid to ecological benefits and realization of the green development of land use, while advancing the economy.

## 5. Conclusions

(1) During the study period, the land use types in Guilin were ranked as follows, according to the area size: woodland, arable land, grassland, waters area, construction land, and unused land. Arable land, woodland, and grassland were concentrated and distributed in a row, and construction land was concentrated in the central region. Urbanization was the main driver of land use change in Guilin, and with the rapid expansion of the city, land use types have changed to different degrees. Among them, the total land mass of arable land decreased the most, whereas the amount of construction land increased the most, and the trend toward change was the most obvious.

(2) Landscape ecological risks in Guilin between 2000 and 2020 can be mostly categorized into lowest-risk, lower-risk, and higher-risk. Significant differences exist in the spatial distribution of landscape ecological risks, with a general trend toward gradual diminishment. In terms of spatial distribution, the highest-risk areas were focused predominantly on the center of the study area, where human activities are frequent, with the risk area gradually decreasing. In contrast, the lowest-risk areas were situated in the northwestern and eastern parts of the study area, as well as the edges of the study area. The overall regional ecological risk class area exhibited a decreasing trend, from the central part to the periphery. 

(3) Woodland and grassland areas were distributed primarily within lowest-risk areas, arable land and waters areas were distributed primarily within higher-risk areas, and construction land and unused land were distributed primarily within higher-risk areas and highest-risk areas. Although the ecological risk levels in the six districts clearly indicated a downward trend, they increased in the counties of Xing’an and Quanzhou, and the areas of the highest-risk expanded. In the process of urban development, it is necessary to formulate corresponding land use policies according to regional development differences.

(4) Moran’s I index of ecological risk in Guilin from 2000 to 2020 exhibited positive results, which were greater than 0.05, indicating a strong autocorrelation. The spatial differentiation of the ecological risk indicated an overall downward trend, while the aggregation exhibited an upward trend. In terms of spatial distribution, the local agglomeration phenomenon was obvious, and the aggregation forms were mainly High-High aggregation and Low-Low aggregation. The High-High aggregation was mainly concentrated in the central, southern, and northeastern regions, which forms a strong spatial aggregation. Low-Low aggregation was mostly concentrated in the northwestern region of the study area, with a small portion in the southern and eastern regions. The marginal aggregation form changed from sporadic to massive.

This study used landscape ecological risk assessment to provide an integrated perspective for the urbanization of landscape resource cities, and its results can provide a reference for land use planning and construction of landscape resource cities. In addition, the study of the ecological risk evolution characteristics at the county scale was helpful to understand the ecological environment of each district and county. It is conducive to the formulation of ecological protection and management policies, in accordance with the economic development characteristics of each district and county. The urban development of Guilin is faced with the contradiction between ecological protection and land use construction, and there is a more urgent need to deal with the ecological risk of the landscape from the perspective of sustainable development.

## Figures and Tables

**Figure 1 ijerph-20-02045-f001:**
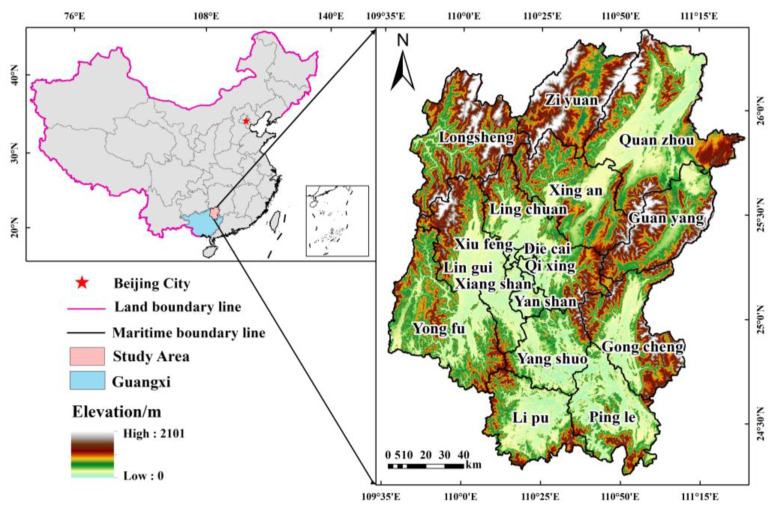
Location of the study area.

**Figure 2 ijerph-20-02045-f002:**
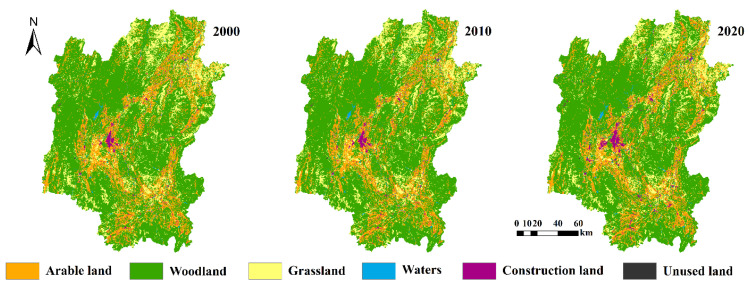
Land use types of Guilin from 2000 to 2020.

**Figure 3 ijerph-20-02045-f003:**
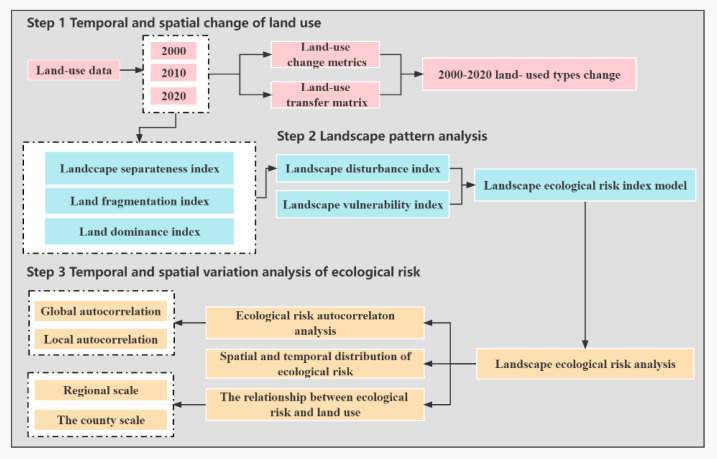
The framework of this study.

**Figure 4 ijerph-20-02045-f004:**
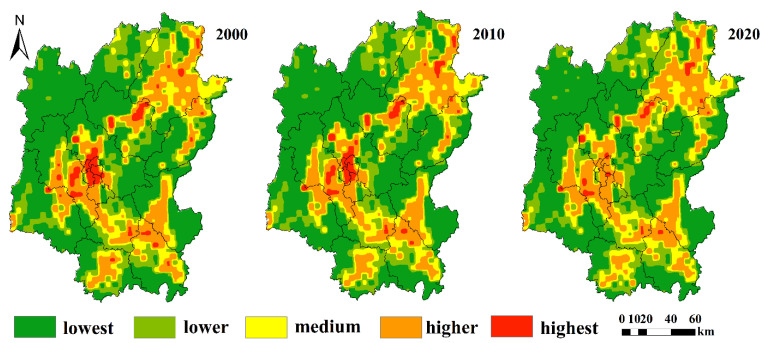
Spatial distribution of ecological risks in Guilin.

**Figure 5 ijerph-20-02045-f005:**
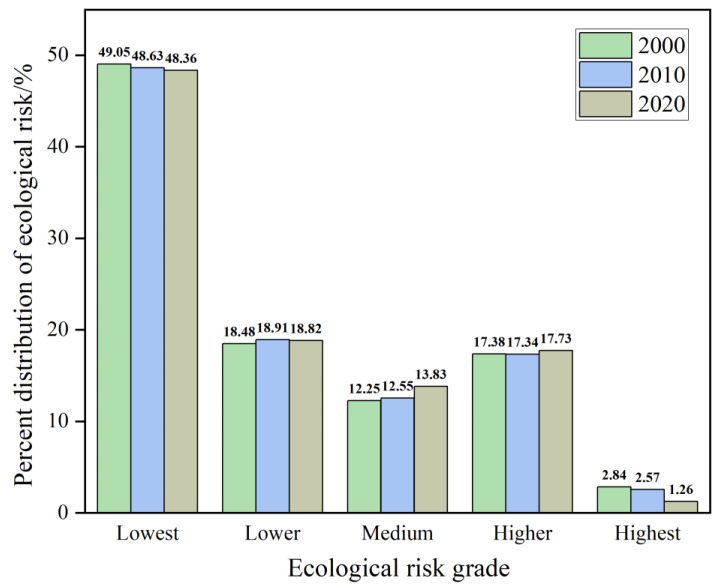
Percentage of land mass at each level of landscape ecological risk in Guilin.

**Figure 6 ijerph-20-02045-f006:**
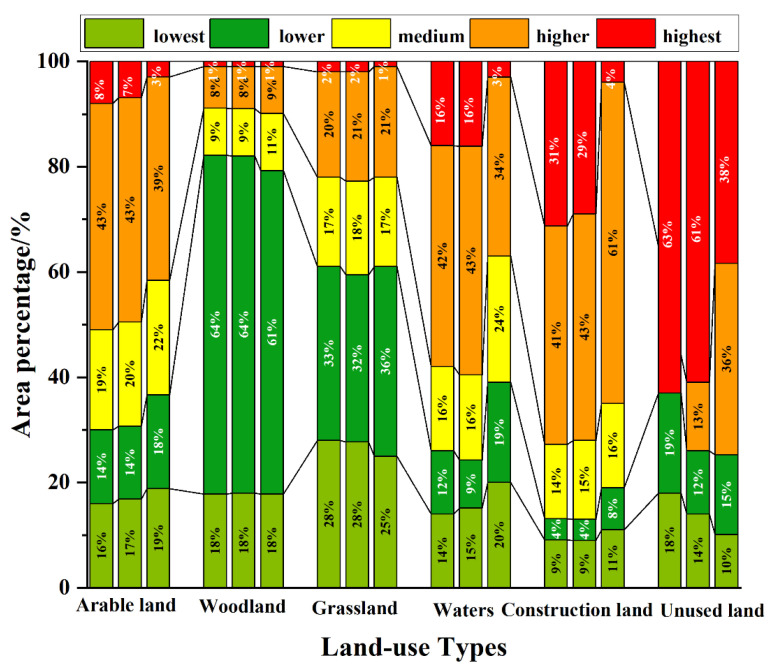
Distribution of ecological risks in different land-use types.

**Figure 7 ijerph-20-02045-f007:**
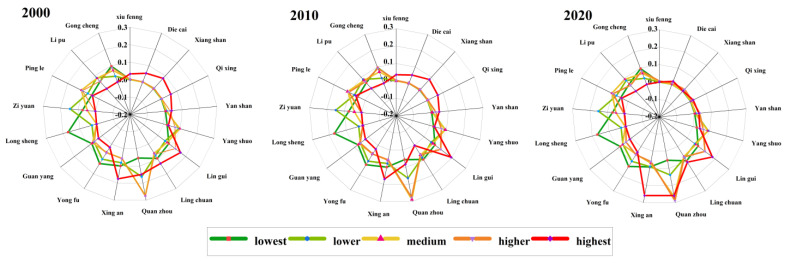
Ecological risk distribution in each district and county.

**Figure 8 ijerph-20-02045-f008:**
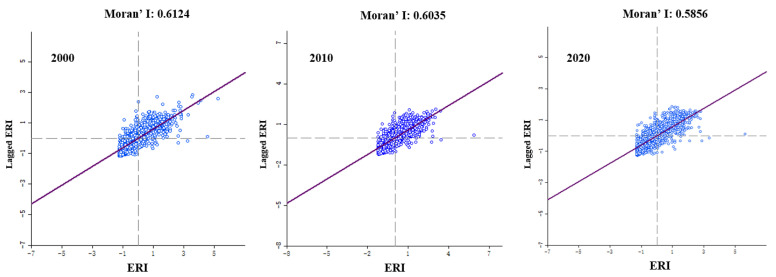
Distribution of the Moran-s I scatter of ecological risk from 2000 to 2020.

**Figure 9 ijerph-20-02045-f009:**
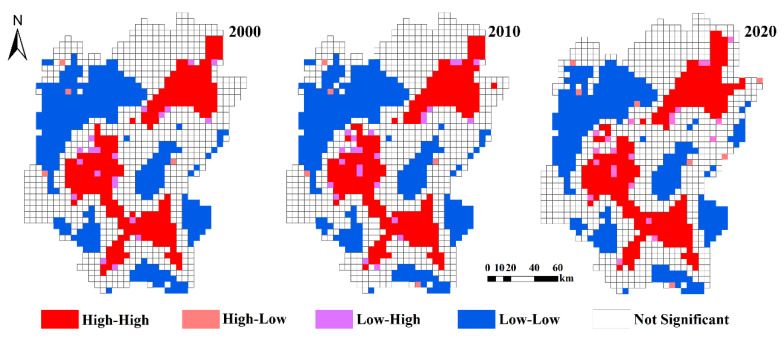
Local autocorrelation diagram of ecological risk.

**Table 1 ijerph-20-02045-t001:** Formula and meaning of the Landscape index calculation.

Index	Formula	Meaning of Index
Landscape fragmentation index ($C_{i}$)	$C_{i}=\frac{n_{i}}{A_{i}}$	Describes the development trend of landscape patches from a centralized continuous state to a discontinuous state as a result of the influence of natural or human factors. The higher the value, the less stable the regional landscape; *ni* indicates the number of patches of the landscape type *i*; and *Ai* is the total area of the landscape type *i*.
Landscape separateness index ($B_{i}$)	$B_{i}=\frac{A}{2A_{i}}\sqrt{\frac{n_{i}}{A}}$	Represents the degree of separation between different patches in landscape types; the separation and spatial distribution of the type of landscapes become more complex and more varied as the value increases [43]; *A* indicates the area of the risk assessment unit.
Landscape dominance index ($D_{i}$)	$D_{i}=\frac{(Q_{i}+M_{i})+2L_{i}}{4}$	This value partially reflects the dominance of a particular landscape type on a regional landscape pattern [44]; *Qi* = the proportion of samples in the evaluation unit *i*/the total number of samples; *Mi* = the number of patches within a certain landscape type *i*/the total number of patches; and *Li* = the area of the landscape type *i*/the total area of samples.
Landscape disturbance index ($N_{i}$)	$N_{i}=aC_{i}+bB_{i}+cD_{i}$	Indicates the level of external disturbances that the local ecosystem has experienced; the higher the disturbance to which a region is exposed, the higher the ecological risk that region suffers [38]; *a*, *b*, and *c* is the weight of the indexes, and *a* + *b* + *c* = 1. According to the findings of the previously conducted research, *a*, *b*, and *c* were assigned the weights of 0.5, 0.3, and 0.2, respectively.
Landscape vulnerability index ($O_{i}$)	According to prior research	The larger the value, the higher the frequency of the occurrence of disturbances in the regional ecosystem by the outside world; based on the results of previous studies [31,45], six levels were categorized according to each of their levels of vulnerability from high to low [46,47]: 6-unused land, 5-waters areas, 4-arable land, 3-grassland, 2-woodland, 1-construction land.

**Table 2 ijerph-20-02045-t002:** The area and changes of land-use types in Guilin between 2000 and 2020.

Land-Use Types	Area/km^2^			Degree of Change/10^−2^	
	2000	2010	2020	2000–2010	2010–2020
Arable land	5380.53	5361.79	5286.03	−0.03	−0.14
Woodland	17,851.41	17,864.33	17,797.18	0.01	−0.04
Grassland	3763.42	3731.91	3713.38	0.08	−0.05
Waters	260.85	273.17	288.61	0.47	0.57
Construction Land	449.65	473.85	617.34	0.54	3.01
Unused land	2.13	3.17	3.83	4.91	2.08

**Table 3 ijerph-20-02045-t003:** Land-use type transfer matrix in Guilin from 2000 to 2010 (unit: km^2^).

2010	2000						
	Arable Land	Woodland	Grassland	Waters	Construction Land	Unused Land	Sum
Arable land	5305.67	35.47	9.82	2.18	8.63	0.00	5361.79
Woodland	30.84	17,785.05	44.44	1.52	2.27	0.02	17,864.33
Grassland	8.95	21.76	3699.44	0.70	1.04	0.00	3731.91
Waters	9.16	3.39	3.91	256.16	0.54	0.00	273.17
Construction Land	25.16	5.66	5.81	0.29	436.94	0.00	473.85
Unused land	0.74	0.07	0.00	0.01	0.24	2.11	3.17
Sum	5380.53	17,851.41	3763.42	260.85	449.66	2.13	27,708.22

**Table 4 ijerph-20-02045-t004:** Land-use type transfer matrix in Guilin from 2010 to 2020 (unit: km^2^).

2020	2010						
	Arable Land	Woodland	Grassland	Waters	Construction Land	Unused Land	Sum
Arable land	4822.77	310.22	80.73	23.69	48.78	0.04	5286.23
Woodland	301.77	17,241.43	228.27	16.03	10.82	0.18	17,798.5
Grassland	83.56	236.81	3381.25	6.19	3.84	0.01	3711.66
Waters	30.33	22.84	8.71	223.59	2.87	0.30	288.64
Construction Land	123.26	50.22	31.71	3.62	407.53	0.00	616.34
Unused land	0.05	1.05	0.03	0.04	0.01	2.64	3.82
Sum	5361.74	17,862.57	3730.7	273.16	473.85	3.17	27,705.19

**Table 5 ijerph-20-02045-t005:** The landscape pattern indices of each land-use type in Guilin from 2000 to 2020.

Type	Year	NP	*C_i_*	*B_i_*	*D_i_*	*N_i_*	*O_i_*
Arable land	2000	5071	0.0094	0.1102	0.3765	0.1131	0.1905
	2010	4984	0.0093	0.1096	0.3765	0.1128	0.1905
	2020	5017	0.0095	0.1122	0.3741	0.1133	0.1905
Woodland	2000	3730	0.0021	0.0285	0.5898	0.1275	0.0952
	2010	3709	0.0021	0.0284	0.5909	0.1277	0.0952
	2020	3715	0.0021	0.0288	0.5873	0.1271	0.0952
Grassland	2000	4294	0.0114	0.1450	0.3463	0.1184	0.1429
	2010	4286	0.0115	0.1460	0.3470	0.1189	0.1429
	2020	4324	0.0117	0.1490	0.3474	0.1200	0.1429
Waters	2000	500	0.0141	0.5234	0.1068	0.1854	0.2381
	2010	447	0.0164	0.6442	0.1076	0.2230	0.2381
	2020	445	0.0151	0.597	0.1121	0.2091	0.2381
Construction Land	2000	5103	0.1438	1.6756	0.2105	0.6167	0.0476
	2010	4840	0.1021	1.2220	0.2096	0.4596	0.0476
	2020	5006	0.0793	0.9355	0.2225	0.3648	0.0476
Unused land	2000	4	0.0188	7.8270	0.0011	2.3577	0.2857
	2010	9	0.0284	7.8882	0.0026	2.3812	0.2857
	2020	10	0.0264	6.9894	0.0028	2.1106	0.2857

**Table 6 ijerph-20-02045-t006:** Statistics on local autocorrelation types of ecological risks in Guilin from 2000 to 2020.

Type	2000	2010	2020
Not Significant	629	629	631
High-High	250	251	258
Low-Low	326	320	311
Low-High	20	24	22
High-Low	4	5	7

## Data Availability

Not applicable.

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
