# Peer review of "Landscape Pattern and Ecological Risk Assessment in Guilin Based on Land Use Change"

_ijerph, 2023, doi:10.3390/ijerph20032045_

Round 1

Reviewer 1 Report

This paper uses multi-period land use data to measure land use changes and ecological risk in Guilin China. The topic is attractive. However, before publishing, the following issues should to be handled.

1. The research question is not well recognized.

2. Question about landscape ecological risk model (subsection 2.3.2).

The indicators for ecological risk is rather arbitrary. Only 5 landscape index without other indicators is not sufficient to measure or cover the ecological risk for specific region.

(lines 152-155) Why is 5 km grid (fishnet)? Whether scale effect has been considered? I have calculated autocorrelation (moral's I) before, and the gird size has a greater impact on the analysis results.

(lines 158-162) What is the criterion for dividing ecological risks into 5 classes?

3. It is inadequate to only analyze the autocorrelation of ecological risk of Gulin in research content. And a driving analysis on ecological risk is recommended.

Reviewer 2 Report

I have gone through the manuscript titled “Landscape Pattern and Ecological Risk Assessment in Guilin

Based on Land Use Change”. In my opinion, despite authors have been successful in reaching their aims and goals, I would like to suggest the followings:

1.      The end part of abstract need more generalization conclusion. The current abstract only emphasizes on local more than general conclusion.

2.      The keywords provided by authors are mainly derived from the main title. Authors should try to provide some different keywords. This would increase the visibility of paper by search engines.

3.      Introduction is need some revision. This part should be of international interest. Therefore, authors should try to present more research works from different parts of world. At the same time, it expected that the literature review determined the research gap which it will approve the novelty of work.

4.      Material and methods are well written. However, I suggest authors The place should introduce in relation to the China then in Gungxi Zhuang area as it presented in Figure 1.

5.      Some information on Quality Control (QC) and Quality Assurance (QA) of the carried analysis seems necessary.

6.      The method of calculation of ecological risk is not clear, while it expected that applied method should be replicable.

7.      The titles of figures are too short. I suggest authors to provide more detailed titles for the figures.

8.      In result, the explanation of reason/s and clarifying the effective factor/s which influence on ERI change seems less considered. Therefore, it needs more consideration.

9.      Authors indicate to industrial and a influential corridor in line 426-429, if these area and corridor are effective, it expected to introduce them in study of area.

1.  The result and discussion part is well written and justified. However, authors should try to compare the obtained results with the previously published papers.

11.  The conclusion need some revision. Normally, conclusion is more than the brief and abstract of finding. While, it should mainly concentrate on the answering the determined gap/s and also new questions which raised due to this study. Moreover, the short indication of implication of this research could be interesting.

Reviewer 3 Report

Notes are in the fail.

Reviewer 4 Report

This is a very interesting and well prepared paper concerning important problem of city / urban region development in relationship with environment quality, with main emphasis on correlation between landscape pattern and Ecological Risk Assessment on the base of Land-Use Change Data - in the period 2000 - 2020 in selected cities in China (the main study area: Guilin and other cities for comparison).

All sections of the paper are generally clearly formulated and well performed. However, the methodical value of the paper is average, in my opinion. Presented methods have been known and developed since a few decades all over the world. However, the way of use of these methods, selection / introduction of parameters, way of elaboration of research results, interpretation of results are partly new /orignal - it will be interesting for many readers researching topics of this nature. The most interesting is evaluating of ecological risk.

Other notes:

Ad DISCUSSION and also RESULTS. In general, the problem of "urban sprawl" is not indicated as a threat (cities should develop, but within minimum occupation of a new area - outside and inside borders of a city; compact building units should be placed in the vast green network, recycling of terrains having lost their previous function - in the cities of 21st century).

Ad RESULTS and CONCLUSIONS. By the way, it is intersting: how it is possible: general level of ecological risk has been decreased in the study area during 20 years, when woodland decreased and construction area increased and fragmentation of landscape increased... - ?

In general: I estimate this paper as interesting and well constructed, concerning important current problems. As regards innovation of issues, I estimate: average level. Methods are known, but detailed methodical elements (selection of parameters, approach to research of presened problem) is partly new, anyway intersting. The paper is worth publishing after minor revisions.

Reviewer 5 Report

1. What are the innovative points in the process of ecological risk assessment in the thesis, besides taking Guilin as the research object for ecological risk assessment? Can you express the novelty of the research method or research content of the thesis in depth in the introduction?

2. As a typical karst landscape area, Guilin has fragile ecological background, soil erosion, rock desertification and other environmental problems, and the conflict between human and land is prominent. Is the research method of using land use data only to calculate the landscape pattern index to reflect the ecological risk of karst landscapes reasonable or more relevant? Does the land use data fully summarize the characteristics of karst landscapes?

3. Why did we choose the 20 years of land use data from 2000, 2010 and 2020 for the spatial and temporal characteristics analysis, is there any special features?

4. Is it reasonable to evaluate the ecological risk of Guilin as a whole without dividing the risk area in the paper? Should the division of ecological risk plots be carried out according to local specificities before evaluation?

5. In paragraph 319, the spatial autocorrelation analysis of the ecological risk index in the region for three years does not explain what causes the ecological risk in the region to show such a spatial layout. Also the ecological risk of landscape under karst landscape is greatly influenced by geological and geomorphological conditions, is there a more reasonable method for spatial analysis?

Round 2

Reviewer 1 Report

Good job!  But, I still think the research question less interest to readers. It is necessary to reorganize this paragraph.

Reviewer 5 Report

1) The response to the innovation point is vague and inadequate. The assessment studies of coupled watershed conservation mentioned in the paper are still relatively numerous.

2) The core research method of the article for land use types reflecting ecological risks of karst landscapes is still not modified, and the evaluation method using landscape pattern index is not comprehensive enough. It is unreasonable to categorize rocky desertification arable land and underground karst caves simply as arable land and unused land, and it is not a complete reflection of the real ecological risk of karst areas without considering other ecological influence factors.
